# A 5Ad Dietary Protocol for Functional Bowel Disorders

**DOI:** 10.3390/nu11081938

**Published:** 2019-08-17

**Authors:** Fandi Ibrahim, Philippa Stribling

**Affiliations:** Division of Life Sciences, School of Engineering, Arts, Science and Technology, University of Suffolk, Ipswich IP4 1QJ, UK

**Keywords:** functional bowel disorders, irritable bowel syndrome, dietary therapies, abdominal pain, bloating, constipation, diarrhoea, IBS, 5Ad Dietary Protocol, FODMAPs

## Abstract

Functional bowel disorders (FBDs) affect around 20% of the population worldwide and are associated with reduced quality of life and high healthcare costs. Dietary therapies are frequently implemented to assist with symptom relief in these individuals, however, there are concerns regarding their complexity, restrictiveness, nutritional adequacy, and effectiveness. Thus, to overcome these limitations, a novel approach, the 5Ad Dietary Protocol, was designed and tested for its efficacy in reducing the severity of a range of gastrointestinal symptoms in 22 subjects with FBDs. The protocol was evaluated in a repeated measures MANOVA design (baseline week and intervention week). Measures of stool consistency and frequency were subtyped based on the subject baseline status. Significant improvements were seen in all abdominal symptom measures (*p* < 0.01). The effect was independent of body mass index (BMI), age, gender, physical activity level, and whether or not the subjects were formally diagnosed with irritable bowel syndrome (IBS) prior to participation. Stool consistency and frequency also improved in the respective contrasting subtypes. The 5Ad Dietary Protocol proved to be a promising universal approach for varying forms and severities of FBDs. The present study paves the way for future research encompassing a longer study duration and the exploration of underlying physiological mechanisms.

## 1. Introduction

Functional bowel disorders (FBDs) are characterised by symptoms of the mid or lower gastrointestinal (GI) tract in the absence of any structural or biochemical abnormalities, and include irritable bowel syndrome (IBS), functional constipation, diarrhoea, bloating, and recently included opioid-induced constipation [1,2]. The most common of these disorders is IBS, which is estimated to have a global prevalence of around 11% [3] and is associated with the presence of recurrent abdominal pain, related to alterations in bowel habits [4]. These symptoms must be present for at least one day per week for the last three months, with symptom onset occurring at least six months prior to diagnosis, to meet the Rome IV diagnostic criteria. Additionally, based on the individual’s predominant stool patterns, IBS is categorised into the following subtypes: IBS with predominant constipation (IBS-C), IBS with predominant diarrhoea (IBS-D), IBS with mixed bowel habits (IBS-M), and IBS unclassified (IBS-U) [5], IBS-C and IBS-D are believed to each account for one third of IBS cases [6]. IBS has been shown to affect more women than men, with women displaying more constipation-related symptoms, and men more diarrhoea-related symptoms [7,8]. Other risk factors for IBS include younger age and preceding gastrointestinal infections [6]. Between 6%–17% of individuals experience post-infectious IBS (PI-IBS) after suffering from an acute episode of infectious gastroenteritis [9]. Severe IBS symptoms have been linked to greater levels of depression, impaired physical functioning, and lower quality of life for the individual [2]. Furthermore, IBS has a great impact on society, with overall healthcare costs per year of around £1.2 billion [10]. It is also estimated that those with IBS are twice as likely to take time off work than healthy individuals [11]. Functional constipation, diarrhoea, and bloating are diagnosed based on insufficient criteria for IBS diagnosis [12], however, there is great difficulty in differentiating between each FBD, such as IBS-C and functional constipation, due to the significant symptom overlap and change in intensity and severity of symptoms over time [13,14,15,16]. It is also difficult to determine the prevalence of each FBD due to differences in the definitions used, with self-reported rates of chronic constipation, for example, being higher than more objective measures using the Rome criteria.

Food and dietary factors play a major role in the pathogenesis of symptoms in FBDs and are proposed to interact with our intestinal and colonic epithelia via the following: immune-mediated reactions, stimulation of the enteric nervous system (ENS) through chemical stimuli, luminal distention through mechanical forces, and unknown mechanisms [17]. The clear involvement of dietary factors in symptom generation in some individuals has been demonstrated by periods of fasting, which have been shown to reduce symptoms of abdominal pain and discomfort, bloating, and diarrhoea in those with IBS [18].

There is no cure for FBDs, therefore, therapies are targeted at symptom reduction and, the predominant therapies include drug treatment, nutrition, and psychotherapy [6]. Dietary therapies are now frequently implemented in place of medication due to the high association between symptom onset and the ingestion of certain foods [19,20]. A few dietary approaches have been utilised to address FBDs such as the low food chemical diet/elimination diet, the low amine/histamine diet, the low capsaicin diet, the gluten-free diet, and recently the low fermentable oligo-, di-, mono-saccharides and polyols (FODMAP) diet [20,21]. However, there is a lack of high-quality evidence surrounding their efficacy, and the existing dietary approaches have raised concerns regarding their complexity, restrictiveness, nutritional adequacy, and effectiveness, presenting an evident need for a new dietary approach for the management of FBDs.

Therefore, a novel approach, the 5Ad Dietary Protocol, was designed to fill a gap in the existing research and to overcome the limitations associated with existing dietary approaches regarding complexity, restrictiveness, nutritional adequacy, and effectiveness (Box 1). During the design of both the 5Ad diet and its post-intervention guidance (the full 5Ad Dietary Protocol, Appendix A), utmost care was taken to ensure the nutritional adequacy of the protocol. The nutritional adequacy was evaluated using Nutritics v5.094, employing COFIDS 2015 as the database, and the average weekly intake of the nutrients were assessed against the DRVs/RNIs as set by the SACN/COMA guidelines, UK, 2017 (Appendix A).

Based on the nature of the diet (Box 2), it was hypothesized that the 5Ad Dietary Protocol would be effective in reducing abdominal symptoms in all FBDs and would also improve stool consistency and frequency in both those with diarrhoea and those with constipation.

Box 1Aims of the 5Ad Dietary Protocol.
√Provide a dietary approach that is
less restrictive, uncomplex, nutritionally adequate, and suitable for
long-term adherence.√Produce a universal dietary approach for functional bowel
disorders (FBDs), effective in reducing, not only abdominal symptoms, but the
wide range of complaints associated with all forms and severities of FBDs,
including diarrhoea- and constipation-related symptoms.√Remove the need to distinguish between each FBD when
considering dietary management for symptom relief.


Box 2Concepts of the 5Ad Dietary Protocol *. * Meal examples are provided in Appendix A.
√Adopt
a “bottom-up” approach to exclude only foods with a large amount of known offending food components (e.g., oligosaccharides, resistant proteins, food
additives, and highly processed/refined foods).√Use of five simple food
groups from which at least one item should be consumed per day to ensure a
balanced and complementary diet.√Advise the consumption
of around 1 kg fruit/vegetables per day to ensure adequate dietary fibre
intake.√Include only fruits
with equimolar concentration of fructose and glucose.√Include low-lactose
dairy products to ensure adequate calcium intakes.√Focus on foods which
can be consumed rather than a list of foods to avoid.√Use all-natural whole
foods to avoid the need to purchase unhealthy and costly commercial
gluten-free or low fermentable
oligo-, di-, mono-saccharides and polyols (FODMAP) alternatives.√Provide a universal
approach for all forms and severities of functional bowel symptoms, not only
those meeting the irritable bowel syndrome (IBS) diagnostic criteria, to
remove the difficulties associated with distinguishing between each functional
bowel disorder (FBD).√Provide an
easy-to-administer, simple-to-follow, and long-term approach.


## 2. Population and Study Design

Participants meeting the eligibility criteria shown in Box 3 were included in the present study and were recruited via online forums, social media, email, flyers, and bulletin boards in local surgeries, shops, and sports and community centres. The study protocol was approved by the Ethics Committee of the University of Suffolk, UK. The study design and procedures of the study were carried out in accordance with the principles of the Declaration of Helsinki. Participants received written information about the study, completed a screening questionnaire and gave informed consent.

Box 3Participant Eligibility Criteria.
√18 years or over.√Not pregnant (if applicable).√Experience abdominal pain and at least 2
of the following symptoms: chronic constipation, diarrhoea, or an alternation
of both, abdominal bloating, flatulence/excessive gas, bowel urgency,
straining, or incomplete defecation.√These symptoms must be present for ≥3
times per week, with symptom onset occurring ≥1 year prior to participation.√No known underlying pathology (e.g.,
Crohn’s disease, ulcerative colitis, celiac disease), self-reported.√No gastrointestinal surgery within the
past year.√Non-vegans (vegetarians and pescatarians
included if consume eggs and/or dairy).√Those taking prescribed medications which
may affect bowel function included if the intake is maintained throughout the
baseline and intervention period.


A repeated measures design was used to compare baseline functional bowel symptoms experienced over 7 days, whilst participants followed their habitual diet, to symptoms experienced during a 7-day intervention period of adhering to the 5Ad Dietary Protocol. Those eligible for the study were asked to complete a baseline evaluation questionnaire over the 7 days whilst following their usual diet. Upon completion of the baseline period, participants were given a copy of the 5Ad Dietary Protocol including detailed instructions for following the approach, and an intervention evaluation questionnaire to be completed in the same way as the baseline questionnaire, whilst following the protocol for the 7 days (Figure 1). Email was the main method of delivering the forms and returning the completed forms, telephone conversations were also used in response to some clients’ questions, and face-to-face meetings occurred for the participants living nearby. Two participants used postal mail for the return of their completed forms.

Subjects recorded daily bowel movements in the provided questionnaires over the baseline and intervention periods, which included stool frequency and stool type (according to the Bristol Stool Scale). A range of abdominal symptoms (i.e., abdominal pain, bloating, flatulence, urgency, straining, and incomplete defecation) were recorded daily using a 4-point Likert scale coded from 0–10 for conversion into continuous variables. Outcome measures included the cumulative weekly scores of a range of abdominal symptoms, Bristol Stool Scale Score (BSSS), and weekly frequency of defecation. Predictors of response included the effect of age, gender, body mass index (BMI), physical activity level (PAL), and IBS diagnosis.

The changes in the multiple outcome measures (abdominal symptoms) indicated above were statistically analysed using repeated measures MANOVA and the assumptions of residual normality and equality of variances were confirmed through the relevant quantile plots. The BSSS and the weekly frequency of defecation were analysed using between-group repeated measures ANOVA to adjust for the baseline subtypes to give a meaningful interpretation. All statistical analyses were performed using IBM SPSS statistical package, version 25. Any favourable changes in outcome measures, or effect of predictors of response, reaching statistical significance (*p* < 0.05) were regarded as a substantial improvement or an interaction effect, respectively.

## 3. Results

### 3.1. Participants and Baseline Characteristics

The participants were recruited using a rigorous screening questionnaire and an 11-page food frequency questionnaire (FFQ) to ensure that they were not following any specific medical diets (e.g., diabetic) or the low FODMAP diet and that they did not have any health issues that we would need to be concerned about during the intervention. Those who were using non-prescribed medication/supplements were eligible only if they agreed to keep the same therapy during the baseline as well as the intervention phase. One participant was taking a prescribed medication, and she obtained permission from her GP to participate in the study. Apart from this one participant, none of the other participants were following a specific diet nor had medical issues that would require further consultation. Based on the FFQs and screening questionnaires, all participants had been consuming variations of common foods. Additionally, there were a few cases of self-avoidance of mainly milk and wheat products. All 38 of the subjects screened for the study were deemed eligible to take part. Of these, 5 did not start the baseline period for unspecified reasons, and 33 completed the baseline period. A total of 9 subjects did not start the intervention period for unspecified reasons and 24 completed the intervention period. One subject was excluded based on self-reported non-compliance and one due to unrelated underlying health issues, leaving 22 subjects (16 women (73%)) included in the analysis (Figure 2). Baseline characteristics are shown in Table 1.

### 3.2. Outcome Measures

All abdominal symptom scores were significantly decreased from the baseline week to the intervention week as follows (Figure 3): abdominal pain (24.26 ± 4.11 versus 10.59 ± 2.44, *p* = 0.0001), bloating (31.73 ± 3.36 versus 11.91 ± 2.66, *p* = 0.0001), flatulence (29.46 ± 3.52 versus 13.18 ± 1.90, *p* = 0.0001), bowel urgency (19.59 ± 3.90 versus 10.55 ± 2.49, *p* = 0.006), straining (17.50 ± 3.98 versus 8.36 ± 2.08, *p* = 0.009), and incomplete defecation (22.05 ± 3.62 versus 9.68 ± 2.64, *p* = 0.0001). The BSSS (Figure 4a) was significantly increased in the baseline constipated stool type group (1.52 ± 0.28 versus 3.03 ± 0.42, *p* = 0.003) from the baseline to the dietary intervention week. For the baseline diarrhoea stool type group, BSSS was decreased but was not significant (5.42 ± 0.22 versus 5.01 ± 0.25, *p* = 0.070) and normal stool type remained constant (3.65 ± 0.20 versus 3.52 ± 0.27, *p* = 0.751). Weekly frequency of defecation (Figure 4b) significantly decreased for the high baseline stool frequency group (20.17 ± 2.65 versus 13.17 ± 1.88, *p* = 0.0001), did not change significantly for the normal baseline stool frequency group (7.60 ± 0.40 versus 8.30 ± 0.93, *p* = 0.405), and marginally increased for the low baseline stool frequency group (3.60 ± 0.68 versus 9.20 ± 3.15, *p* = 0.096). The significant improvements shown in all abdominal symptoms, BSSS, and weekly frequency of defecation during the dietary intervention relative to baseline occurred irrespective of baseline characteristics, as there was no significant effect of gender, age, BMI, PAL, or IBS diagnosis on any of the outcome variables.

## 4. Discussion

The 5Ad Dietary Protocol showed promising findings regarding its efficacy in reducing a wide range of symptoms associated with FBDs. A significant improvement was shown in abdominal pain as well as all other assessed abdominal symptom scores. Although not experienced in all subtypes, significant improvements were seen in stool consistency for those with baseline constipation, and in weekly frequency of defecation for those with a high baseline weekly stool frequency. The remaining subtypes also showed a trend towards normalisation of bowel habits. The fundamental aim of the study was to design to a simple-to-administer and simple-to-follow dietary approach which is nutritionally adequate and suitable in the long-term. The significant enhancement of bowel function demonstrated in this present study should encourage further research into the effects of long-term adherence to the 5Ad Dietary Protocol. Gastrointestinal transit time is normally up to 72 h [22], and therefore, we presumed that the one-week duration of the intervention would be adequate to observe any significant changes in bowel habits and the associated symptoms. Indeed, when comparing the average results of the last four days of the intervention week to the first three days of the same week (72 h), further significant improvements could be distinguished within the same week. However, this sub-analysis was not reported in this work for consistency in comparing to the baseline week and to avoid any bias in the selection of days.

The way in which the 5Ad Dietary Protocol differed from the complexity of the low FODMAP diet was by adopting the concept of consuming at least one item of food from each of the five food groups every day. In addition to providing simplicity, it also ensured that a balanced diet was followed, unlike the low FODMAP and gluten-free diet, which do not prioritise healthy eating, therefore increasing the risk of nutrient deficiencies [23,24,25]. In terms of being less restrictive, the 5Ad Dietary Protocol adopted more of a “bottom-up approach”, whereby only foods with very large amounts of FODMAPs or one particular FODMAP were excluded, in contrast to the “top-down approach” of the low FODMAP diet itself, whereby over-restriction occurs of all or most FODMAPs [21]. Furthermore, the approach proved easier and less time-consuming to administer than the low FODMAP diet [26].

Because the 5Ad Protocol is a new dietary approach and was not compared to an active intervention in this present study, it is difficult to compare its efficacy to existing dietary approaches for FBDs. In fact, making comparisons between existing research regarding the low FODMAP diet itself is a challenge due to the great heterogeneity between study designs, methods of symptom assessment, inclusion criteria/study population, primary and secondary outcomes, and the use of another dietary intervention or control group [27]. The comparison of the low FODMAP diet to participants’ habitual diet [28] and to a “typical” Australian diet [29] are the closest study designs to that of the 5Ad Protocol regarding the concept of comparing an active intervention, designed to improve symptoms, to a standard or typical diet, and therefore provide a reasonable basis for comparison. Additionally, the 5Ad Dietary Protocol is low in specific oligosaccharides and will, therefore, be similar to the low FODMAP diet in some important respects. However, we do not assume that all FODMAPs are the same; some oligosaccharides have important prebiotic effects (e.g., human/bovine milk oligosaccharides), while others cause intestinal hypersensitivity (e.g., raffinose and stachyose, common in legumes) for people with FBDs. The same applies for disaccharides (e.g., lactose versus maltose) and polyols (e.g., erythritol versus sorbitol) as these have large differences in bioavailability and fermentability by gut microbes. We also excluded some products based on their resistant protein and food additive contents.

In line with most research regarding the low FODMAP diet, abdominal pain, bloating, and flatulence were the most significantly improved abdominal symptoms in this present study. These are undoubtedly the most troublesome and frequent symptoms among all IBS subtypes [30], with abdominal bloating occurring in 96% of those with IBS [12]. However, an aspect of the protocol was also responsible for normalising bowel function in both those with constipation and diarrhoea, which is not a known feature of the low FODMAP diet [30]. Whilst the reductions in these abdominal symptoms are likely to have been influenced by the removal of some FODMAPs and/or gluten, the exclusion of additional food components addressed by the 5Ad Protocol, which are poorly studied in relation to FBDs, are also suspected to have contributed to symptom improvement (e.g., plant resistant proteins and additives such as carboxymethyl cellulose, carrageenan, etc.).

It is suggested that it is the visceral hypersensitivity to luminal distention, produced by excessive gas, which causes sensations of bloating and abdominal pain/discomfort in those with IBS, rather than the level of gas production compared to that of healthy controls [31]. It is plausible that the 5Ad Protocol led to reductions in gas production, therefore resulting in decreased flatulence, but also to reduced luminal distention and therefore lower levels of bloating and abdominal pain.

Previous studies comparing the low FODMAP diet to the traditional dietary advice showed inconsistencies regarding faecal indices in all IBS subtypes. Constipation was the least improved IBS symptom during both approaches; additionally, only a slight trend was witnessed for improved diarrhoea in the low FODMAP group by Staudacher et al. [32] and only traditional advice resulted in significant improvements regarding bowel habit dissatisfaction in a study by Böhn et al. [27]. Furthermore, the number of bowel movements were significantly reduced during the low FODMAP diet, but mean stool consistency was unchanged for both approaches [27]. These findings suggest that constipation and/or stool consistency in all subtypes are rarely improved by first- or second-line dietary advice for IBS, and only slight improvements are apparent in those with diarrhoea during adherence to a low FODMAP diet. The 5Ad Protocol demonstrated a benefit for both those with constipation and diarrhoea in terms of stool frequency and consistency.

Normal defecation frequency is classed as experiencing between three bowel movements per week to three per day [33]. In this present study, six to ten bowel movements per week was considered a more appropriate estimation of normal/desirable bowel function. Therefore, weekly stool frequency was categorized into the following baseline subgroups: low (<6), medium (6–10), and high (>10). Defecation frequency, however, has in fact been shown to be unrelated to intestinal transit time or daily faecal weight [33], which are both measures of intestinal and digestive health [34]. IBS subtype classification, however, is based largely on predominant stool form/consistency [5] which, based on the validated Bristol Stool Scale, is suggested to be well correlated with transit time and faecal output [33,35].

This present study also assessed measures of bowel urgency, straining, and incomplete defecation, which were all significantly improved from baseline. These symptoms, however, may not be reflective of stool form/consistency or, therefore, transit time. Occurring in around 72% of those with self-reported diarrhoea, bowel urgency has been described as the most bothersome symptom in those with IBS-D [36]. Urgency is generally caused by the presence of watery stool, as a result of rapid transit, which is hard to retain [12,36,37], however, a sense of urgency with frequent defecation but featuring solid stool can also occur, known as pseudodiarrhoea [12]. Additionally, straining and incomplete defecation are commonly associated with constipation but, straining, which can be present due to feelings of incomplete defecation, may also occur with soft/watery stools [12].

The above factors further support the concept that the measure of stool form/consistency is a more reliable indicator of transit time, and therefore bowel function and health, than stool frequency. In terms of stool consistency, like in the previously-mentioned study by Staudacher et al. [28], there was no significant difference in stool type for those with diarrhoea after following the intervention diet from the habitual diet. However, there was an improvement in stool type towards normal/ideal, with Staudacher and co-workers [28] also witnessing a greater percentage of normal/ideal stool type in the low FODMAP diet group. For those with constipation following the 5Ad Protocol, a significant positive change was witnessed in stool type from baseline. The findings suggest that the 5Ad Protocol is effective at normalising stool form/consistency in both those with constipation and those with diarrhoea. The exclusion of those with constipation in the study by Staudacher and co-workers [28], however, does not allow for comparisons in this respect. Regardless of which stool type is reflected by urgency, straining, and incomplete defecation, the significant improvements in these symptoms in this present study demonstrated the all-round effectiveness of the 5Ad Protocol.

As detailed earlier, the 5Ad Protocol was assessed for nutritional adequacy and was designed to be suitable to follow in the long term. Lactose restriction during the low FODMAP diet is associated with reduced calcium intake due to avoidance of dairy products, increasing the risk of vitamin D deficiency [25]. Up to 82% of those with IBS-D with self-reported lactose intolerance were shown to tolerate 10 g lactose (equivalent to one cup of milk), demonstrating that some individuals avoid dairy products unnecessarily. The 5Ad Protocol included small amounts of milk (if required in tea), plain natural yoghurt, and mature cheese, to ensure adequate calcium intakes, whilst keeping lactose intakes low [26].

Additionally, due to the restriction of wheat, rye and barley during the low FODMAP and gluten-free diet, and legumes, and some fruit and vegetables during the low FODMAP diet [21,26], both approaches have been associated with reduced dietary fibre intake [23,24]. Dietary fibre accelerates intestinal transit and promotes laxation through contribution to faecal weight [29]. Although a controversial topic, lack of dietary fibre has long been associated with increased risk of constipation, therefore, increasing dietary intake has traditionally been recommended to those with IBS and chronic constipation [38]. The removal of all cereals and legumes during the 5Ad Protocol may raise similar concerns, however, fibre from fruits and vegetables has been shown to produce similar effects as fibre from cereals, promoting regularity through significantly increased faecal weight and decreased intestinal transit time [34]. Thus, participants were encouraged to consume 1 kg of fruit/vegetables per day to ensure adequate dietary fibre intakes during this present study. This itself may have contributed to the improvements in constipation, considering that many individuals fail to meet the recommended dietary fibre intake of 30 g per day [39].

Due to their prebiotic effects, the concerns surrounding oligosaccharide restriction are more likely related to the subsequent effect on the microbiome and SCFA production [40] although this effect has not always been observed [24], and some studies have shown increases in beneficial bacteria, such as Actinobacteria [41]. Changes to the microbiome may also feature during the 5Ad Protocol due to the exclusion of oligosaccharides, however, with the high intake of fruit, vegetables, and nuts, but these changes are likely to be positive in terms of gut microbial composition and metabolites, which needs to be substantiated in further studies. It is also worth considering that many individuals with IBS display dysbiotic microbiome before engaging in a dietary intervention [42,43] and, as discussed previously, the Western diet is strongly associated with causing dysbiosis, in addition to its further damaging effects [44]. In fact, it has been suggested that changes to the gut microbiota reported during adherence to a gluten-free diet may be due to the consumption of unhealthy gluten-free products rather than the removal of gluten-containing cereals [20].

The lack of significant interaction of predictors of response (gender, age, BMI, PAL, and IBS diagnosis) and outcome measures indicates that the 5Ad Protocol was effective in all individuals, regardless of their baseline characteristics, adding further strength to its suitability as a universal approach, but this is awaiting validation in a randomized clinical trial. A limitation with most research in this field is that there are no validated methods of objectively measuring predominant IBS symptoms, such as abdominal pain and bloating [30]. Therefore, most methods of symptom assessment used are subjective patient-reported measurements, which are likely to produce high inter-individual variation [45]. However, the repeated measures design of this present study reduces this risk [46]. Additionally, FBDs are associated with a high placebo response [47]. In the present study, participants were aware that the protocol had been designed to improve symptoms. Therefore, a placebo effect may have been present. However, the highly significant improvements witnessed in all measured variables, using MANOVA testing, suggest that the placebo effect is unlikely to be of significant impact. Additionally, highly appreciated qualitative participant feedback was received after a month follow-up, and a sham diet study is planned for further substantiation. Furthermore, the significant findings of this present study were observed in an intervention period of just one week, in comparison to the several-week intervention period practiced during the low FODMAP diet or other dietary interventions, advocating the efficacy of the 5Ad Protocol [48]. Finally, a small sample size can reduce the power to detect a significant effect [46,49], however, considering the small sample size used in this present study (22 participants), the significant improvements shown in all abdominal symptom scores add further to the proof of efficacy of the 5Ad Dietary Protocol.

## 5. Conclusions

The 5Ad Dietary Protocol has proven to be a promising universal approach for varying forms and severities of FBDs. Clearly, an aspect of the protocol was responsible for providing a dichotomous effect, normalising bowel function in both those with constipation and those with diarrhoea, in addition to significantly improving a wide range of functional abdominal symptoms. Thus, the encouraging results of the present study give potential for the 5Ad Dietary Protocol to be used as a new method of dietary management for those with FBDs. The 5Ad Dietary Protocol has demonstrated that the restrictiveness and complexity of the low FODMAP diet can be overcome and provides an approach which is nutritionally adequate and suitable for potential long-term adherence. The memorable food groups and easy-to-follow instructions of the 5Ad Dietary Protocol create a less daunting concept for individuals to follow, removing the complexity and confusion that are associated with the existing dietary approaches. However, since this research was carried out to explore the utility of the newly developed 5Ad Dietary Protocol, further studies, particularly randomized clinical trials, are required to substantiate these findings and to investigate the long-term adherence and the impact on the quality of life of those who adhered to the protocol in the long term.

## Figures and Tables

**Figure 1 nutrients-11-01938-f001:**
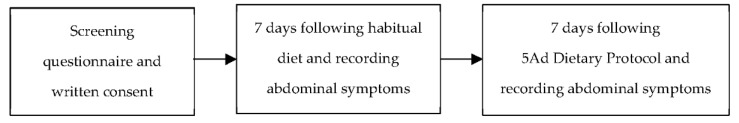
Schematic drawing of the 5Ad Dietary Protocol study design.

**Figure 2 nutrients-11-01938-f002:**
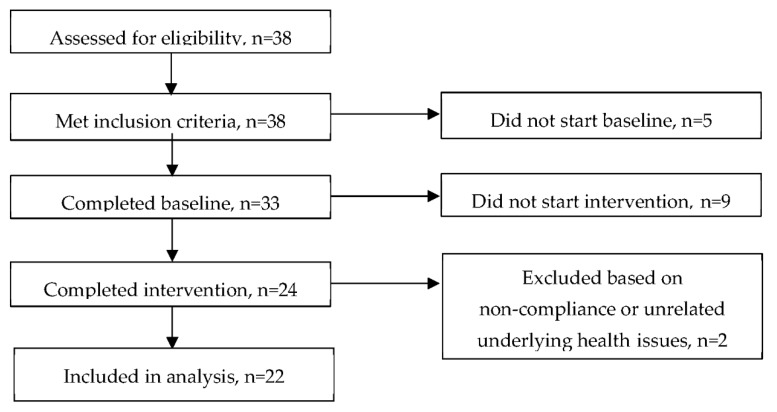
Consort diagram of the number of participants involved in the different phases of the 5Ad Dietary Protocol intervention trial.

**Figure 3 nutrients-11-01938-f003:**
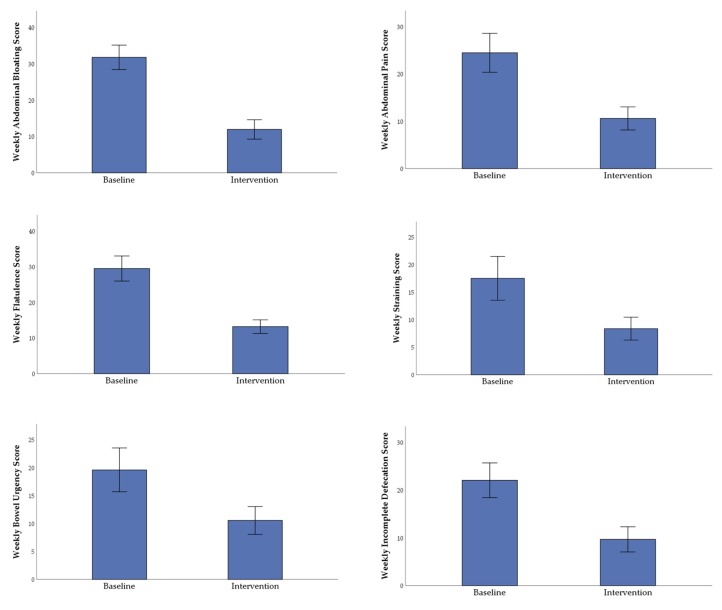
The mean (± SE, *n* = 22) weekly cumulative abdominal symptom scores after 7-day baseline period (participants’ habitual diet) and 7-day intervention period (adherence to 5Ad Dietary Protocol). SE represents standard error; all variables showed a significant difference from baseline to intervention (*p* < 0.01) using repeated measures MANOVA testing.

**Figure 4 nutrients-11-01938-f004:**
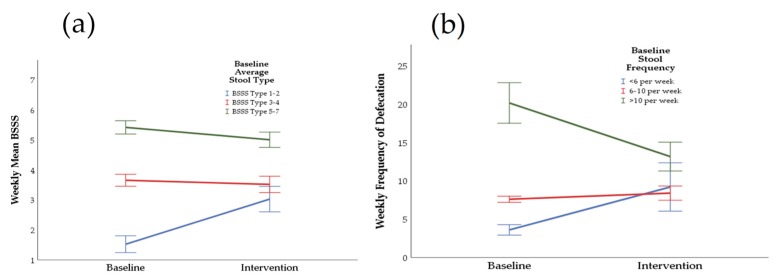
The changes in weekly stool type (BSSS) (**a**) and the frequency of defecation (**b**) as categorized by the baseline status after 7-day baseline period (participants’ habitual diet) and 7-day intervention period (adherence to 5Ad Dietary Protocol). The data show either normalisation or a trend toward normalization of both the stool type and frequency of defecation (*n* = 22).

**Table 1 nutrients-11-01938-t001:** Baseline characteristics of the participants included in the 5Ad Dietary Protocol intervention trial.

Baseline Characteristics	
Age range (mean ± SE)	20–75 (49.29 ± 3.35)
Gender *n*, (%)	
Females	16 (73%)
Males	6 (27%)
BMI category *n*, (%)	
<18.5 kg/m²	1 (5%)
18.5–24.9 kg/m²	13 (59%)
25–29.9 kg/m²	4 (18%)
≥30 kg/m²	4 (18%)
PAL *n*, (%)	
Sedentary	1 (5%)
Light	1 (5%)
Moderate	16 (73%)
Vigorous	4 (18%)
IBS diagnosed *n*, (%)	
Yes	11 (50%)
No	11 (50%)
Baseline average stool type *n*, (%)	
Constipation (type 1–2)	8 (36%)
Normal (type 3–4)	7 (32%)
Diarrhoea (type 5–7)	7 (32%)
Baseline defecation frequency *n*, (%)	
Low (<6 per week)	5 (23%)
Medium (6–10 per week)	5 (23%)
High (>10 per week)	12 (54%)

BMI—body mass index, PAL—physical activity level, IBS—irritable bowel syndrome, SE—standard error

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
