# Peer review of "A 5Ad Dietary Protocol for Functional Bowel Disorders"

_nutrients, 2019, doi:10.3390/nu11081938_

Round 1
Reviewer 1 Report
Manuscript title: "A 5Ad Dietary Protocol for Functional Bowel Disorders ". Author: Fandi Ibrahim and Philippa Stribling Journal: Nutrients I reviewed the above-mentioned manuscript from Ibrahim et al. evaluating the effect of a new dietary protocol on abdominal symptoms in 22 patients with functional bowel disorders. The concept of the 5AD dietary protocol is shows advantages in comparison to the existing low FODMAP diet as it is less restrictive, avoids malnutrition (ensures adequate calcium intake), simple to follow and less costly. Interestingly, this new diet significantly improved abdominal symptoms independent of the IBS subtypes (predominant constipation or diarrhoea). This is a short-term trial studying effects of a 7-day balanced diet in IBS patients consuming at least one item of the 5 food groups, including low-lactose dairy products, around 1 kilo intake of fruit and vegetables, but restricted to fruits with a equimolar fructose to glucose ratio. The authors pointed out three interesting advantages of the diet: - Significant improvement of a wide range of functional abdominal symptoms. - Normalisation of bowel function in predominant constipated and diarrheal patients. - Potential of long-term adherence due to easy to follow instructions. This is a novel and interesting approach towards dietary treatment of FBD and is of interest to the readers of nutrients. The manuscript is well written, scientific and clearly structured. The limitations of the study were clearly presented. Further studies comparing for example 5Ad dietary protocol with low FODMAP are necessary. However, there are just some minor points of criticism that should be referred in the revised manuscript. Minor remarks 1) It is not mentioned how the eligibility criteria “no underlying pathology” was assessed. 2) A more detailed description of the 5Ad Dietary Protocol and how the participants were instructed is missing in the methological section. 3) No information on the diagnosis of the other 50% of the patients were given. 4) Figure 3 and 4 labelling – the letters are to small 5) The introduction could be shortened. 6) The supplement with the meal examples were missing. 7) Pages are missing in the citation of ref 33,48 and 50
Author Response
|
Reviewer: 1 |
|
|
Comment 1 |
I reviewed the above-mentioned manuscript from Ibrahim et al. evaluating the effect of a new dietary protocol on abdominal symptoms in 22 patients with functional bowel disorders. The concept of the 5AD dietary protocol is shows advantages in comparison to the existing low FODMAP diet as it is less restrictive, avoids malnutrition (ensures adequate calcium intake), simple to follow and less costly. Interestingly, this new diet significantly improved abdominal symptoms independent of the IBS subtypes (predominant constipation or diarrhoea). This is a short-term trial studying effects of a 7-day balanced diet in IBS patients consuming at least one item of the 5 food groups, including low-lactose dairy products, around 1 kilo intake of fruit and vegetables, but restricted to fruits with a equimolar fructose to glucose ratio. The authors pointed out three interesting advantages of the diet: - Significant improvement of a wide range of functional abdominal symptoms. - Normalisation of bowel function in predominant constipated and diarrheal patients. - Potential of long-term adherence due to easy to follow instructions. This is a novel and interesting approach towards dietary treatment of FBD and is of interest to the readers of nutrients. |
|
Response |
We fully appreciate the evaluation of the manuscript by the reviewer, and we would like to thank the reviewer for the valuable comments above and emphasizing the novelty of the approach. |
|
Comment 2 |
The manuscript is well written, scientific and clearly structured. The limitations of the study were clearly presented. |
|
Response |
We would like to thank the reviewer for this comment as well. |
|
Comment 3 |
Further studies comparing for example 5Ad dietary protocol with low FODMAP are necessary. |
|
Response |
This original research was carried out to explore the utility of the newly developed 5Ad Dietary Protocol and was intended as a preliminary study; therefore, we do agree with the reviewer that more studies are needed, particularly randomized clinical trials. We planned to conduct an RCT with the 5Ad Dietary Protocol vs the low FODMAP diet as well as the NICE guidelines in collaboration with the regional Clinical Trial Unit (CTU), Norwich, United Kingdom, and we have already established the necessary communication for this purpose. |
|
Comment 4 |
However, there are just some minor points of criticism that should be referred in the revised manuscript. Minor remarks 1) It is not mentioned how the eligibility criteria “no underlying pathology” was assessed. |
|
Response |
A screening questionnaire was the basis of the participant eligibility, and therein the participants were asked whether they have been formally diagnosed with IBS and whether any associated pathologies had been identified. Thus, it was a self-reported assessment, which is now clarified in Box 3. |
|
Comment 5 |
2) A more detailed description of the 5Ad Dietary Protocol and how the participants were instructed is missing in the methological section. |
|
At this point of time, we do feel that there is a potential for intellectual property rights associated with the 5Ad Dietary Protocol and therefore the participants were asked to maintain the confidentiality of the protocol. The 5Ad Dietary Protocol was designed solely by the authors of this manuscript, and although it might prove difficult to patent a dietary protocol, we would like to conduct more studies before we fully publish the details of the protocol. However, if the editor and the reviewers think this is essential before publication, we are happy to upload the detailed instructions of the protocol to share it with the reviewers and the editor and seek their advice on whether the details should be published as supplementary materials, or whether we are able to withhold until investigating the intellectual property matter. |
|
|
Comment 6 |
3) No information on the diagnosis of the other 50% of the patients were given. |
|
Half of the participants were self-reported as formally diagnosed with IBS based on the screening questionnaire and the remaining were not formally diagnosed with IBS. However, the eligibility criteria were more vigorous than the Rome IV criteria and were meant to be more inclusive for those who have been silently suffering from functional bowel disorders rather than focusing only on a limited section of the population who have the opportunity to be diagnosed as IBS patients. The Rome IV criteria applies to the last 6 months before diagnosis, while we included those who have been suffering for at least a year; indeed most participants in this study have been suffering with abdominal symptoms for more than 5 years, according to the screening questionnaire. |
|
|
Comment 7 |
4) Figure 3 and 4 labelling – the letters are to small |
|
The figures have been resized so that the labels are now more visible. |
|
|
Comment 8 |
5) The introduction could be shortened. |
|
We have revisited the introduction and we shortened as recommended. All changes are made using the MS word Review function and can be seen in the new draft. |
|
|
Comment 9 |
6) The supplement with the meal examples were missing. |
|
Apologies for this error, and the example meals are now provided in a file named: Supplementary_1 meal examples. Providing the meal examples may also contribute the discussion of our response to comment 5. |
|
|
Comment 10 |
7) Pages are missing in the citation of ref 33,48 and 50 |
|
The page numbers have now been added to the above citations, and where there was no page number available yet, a DOI has been added. |
|
Reviewer 2 Report
This study was conducted to assess the potential benefit of 5Ad Dietary Protocol i subject with FBDs. Overall it showed promising findings regarding its efficacy in reducing a wide 168 range of symptoms associated with FBDs and a significant improvement was shown in abdominal pain 169 as well as all other assessed abdominal symptoms scores the number of tested subject is of concern. Please provide a justification that n=22 subjects is an optimal group to draw the conclusion.
Author Response
|
Reviewer: 2 |
|
|
Comment 1 |
This study was conducted to assess the potential benefit of 5Ad Dietary Protocol i subject with FBDs. Overall it showed promising findings regarding its efficacy in reducing a wide 168 range of symptoms associated with FBDs and a significant improvement was shown in abdominal pain 169 as well as all other assessed abdominal symptoms scores |
|
Response |
We would like to thank the reviewer for evaluating the manuscript and pointing out the significance of the findings and the improvements shown in a wide range of abdominal symptoms. Indeed, it was very interesting to see these findings and to understand the importance of food ingredients in the aetiology of functional bowel disorders. It is very likely that effects of food ingredients on bowel function are also relevant to extra-intestinal diseases mediated by their direct effect on gut barrier function but this remains to be elucidated in further research. |
|
Comment 2 |
the number of tested subject is of concern. Please provide a justification that n=22 subjects is an optimal group to draw the conclusion.
|
|
Response |
Mathematically, the larger the sample size, the higher the potential to find a significant difference and therefore, a larger sample size is normally required to make sure that the study is not underpowered and to avoid type II error. However, these highly significant findings were observed with a relatively small sample size, therefore demonstrating substantial efficacy of the protocol. From this study, we have also learned about the power required for further randomized clinical trials in order to test the protocol against the mainstream dietary therapies such as the low FODMAP diet and NICE guidelines. We have also developed the methodology for an inclusive evaluation of the abdominal symptoms, and we are working on a novel overall index too. |
Round 2
Reviewer 2 Report
The authors addressed and responded to all questions raised during the first evaluation of the manuscript.
Author Response
We would like to thank the reviewer for this confirmatory statement.